# Balanced Low-Rank Adaptation: Removing Invariance for Fast and Stable Fine-Tuning

## Abstract

Low-Rank Adaptation (LoRA) is the most widely used method for fine-tuning large language models. LoRA is overparameterized: multiple pairs of low-rank factors correspond to the same adapted weight matrix. We observe both theoretically and numerically that these pairs can have significantly different condition numbers: converging to different minimizers of the loss affects the convergence rate of LoRA. Building on this remark, we introduce Balanced Low-Rank Adaptation (BaLoRA), a variant of LoRA that projects iterates onto a balanced manifold where the conditioning of the loss is improved while keeping the same adapted matrix. This projection step is computationally inexpensive and integrates seamlessly with existing fine-tuning pipelines. Empirically, BaLoRA converges faster than standard LoRA and exhibits greater robustness to hyperparameter choices across a range of fine-tuning tasks.

## 1 Introduction

Pretrained foundation models have become indispensable in domains such as natural language processing (Brown et al. (2020); Qin et al. (2023); Taori et al. (2023)), computer vision (Awais et al. (2025)), and multi-modal learning (Li et al. (2022); Liu et al. (2023a)), thanks to their ability to generalize from large-scale training data. These models are generalists in the sense that they have been trained on data with diverse semantic and stylistic content, and they have enough parameters to model a wide range of topics. As such, they are excellent starting points for fine-tuning: when one wants to obtain a model that is a specialist for a certain task for which we have enough data, it is judicious to train on this dataset starting from the pre-trained model. However, as model sizes continue to increase, adapting these models to specific downstream applications poses a major challenge: fine-tuning the full model demands prohibitive computational and storage resources.

To address this issue, parameter-efficient fine-tuning (PEFT) methods have gained significant traction (Houlsby et al. (2019)). These approaches adapt large pretrained models by updating only a small fraction of parameters, drastically reducing cost while maintaining strong performance. Among PEFT techniques, Low-Rank Adaptation (LoRA, Hu et al. (2022)) has emerged as one of the most effective, and builds on the observation that over-parameterized models often have much lower intrinsic dimensionality (Li et al. (2018); Aghajanyan et al. (2021)). Instead of updating dense weight matrices, LoRA introduces trainable low-rank matrices that are added to frozen (*i.e.*, kept fixed) pretrained weights. More precisely, a pretrained weight $W \in \mathbb{R}^{n \times m}$ is updated as $W + AB$, where $A \in \mathbb{R}^{n \times r}$, $B \in \mathbb{R}^{r \times m}$ and $r \ll \min(n, m)$ is the LoRA rank. Since $W$ is frozen, this reduces the number of trainable parameters from $n \times m$ (full fine-tuning) to $r \times (n + m)$, yielding substantial memory savings while preserving adaptability.

Given its success across a broad range of applications, LoRA has recently attracted theoretical interest, with relatively few results available to date. Existing work has investigated its expressivity for feedforward neural networks and transformers (Zeng & Lee, 2024), its fine-tuning dynamics in the Neural Tangent Kernel (NTK) regime (Malladi et al., 2023; Jang et al., 2024), the asymmetric roles of the $A$ and $B$ matrices (Zhu et al., 2024; Hayou et al., 2024b), and the impact of different initialization strategies (Hayou et al. (2024a); Li et al. (2025)).

In this paper, we aim to better understand the training dynamics of LoRA and analyze the conditioning of the underlying optimization problem. We derive the asymptotic convergence rate of LoRA fine-tuning for one-layer linear networks and provide tight bounds on the condition number of the

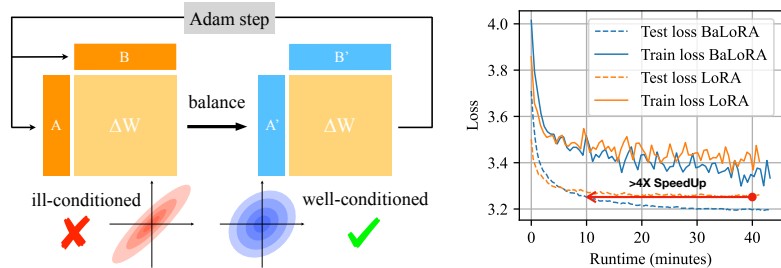

Figure 1: **BaLoRA in a nutshell. Left:** BaLoRA projects the low-rank adapters $(A, B)$ on the balanced manifold after each optimizer step. This projection improves the conditioning of the loss while preserving the product $\Delta W = AB = A'B'$. **Right:** The improved conditioning provided by BaLoRA eases the optimization and therefore accelerates LoRA.

loss at convergence. Central to our study is the fact that LoRA is overparameterized, in the sense that, for any invertible matrix $R \in \mathbb{R}^{r \times r}$, the low rank factors $(AR, R^{-1}B)$ lead to the same adapter. Yet, the optimization landscape is different for all these factors. Our analysis reveals that *balanced minimizers* (*i.e.*, minimizers $(A, B)$ of the loss satisfying $A^\top A = BB^\top$) achieve minimal conditioning. While well-studied in other contexts, *e.g.*, linear networks (Nguegnang et al. (2024)), matrix factorization (Ye & Du, 2021; Ghosh et al., 2025), conservation laws in neural networks (Marcotte et al., 2023), this balanced condition has not been examined for LoRA. Building on our insights, we introduce BaLoRA, an extension of LoRA that enforces balance during training for better conditioning and faster convergence. BaLoRA is lightweight, theoretically motivated, and can be seamlessly integrated with the standard optimization methods.

BaLoRA consists, starting from the standard LoRA initialization, in projecting the low-rank adapters $(A, B)$ on a balanced manifold after each optimizer (*e.g.*, Adam) step (Figure 1). More precisely, letting $AB = U\Sigma V^\top$ the singular value decomposition of the low-rank update at the end of each optimizer step, BaLoRA updates

$$A \leftarrow U\Sigma^{1/2}, \quad B \leftarrow \Sigma^{1/2}V^\top,$$

so that the balanced condition is satisfied by $(A, B)$ at the end of each step. We describe in Section 3 a cheap method to implement this update that does not require computing the SVD of a large matrix. This enforces the adapters $(A, B)$ to converge to a better-conditioned minimizer of the loss, which accelerates asymptotic convergence and may induce better generalization properties (Keskar et al. (2016); Dziugaite & Roy (2017)). In our practical experiments on GPT-2 and Llama-3.2-3B, fine-tuned with the Wikitext-2-raw-v1 dataset, BaLoRA outperforms LoRA and is more robust to changes in the learning rate and initialization scale, with a negligible increase in computational cost. We make the following contributions.

- We introduce BaLoRA, a novel PEFT method that enforces balancedness of low-rank adapters across the optimization with negligible computational overhead. (Section 3)

- We analyze theoretically the conditioning of LoRA's limiting points for a one-layer linear network, proving that balanced minimizers are best conditioned. In that case, BaLoRA therefore converges to a better-conditioned minimizer than LoRA, which increases its asymptotic convergence rate. (Section 2)

- We prove that the GD-BaLoRA iterations can be recast as an intrinsic optimization scheme on the product $AB$. (Section 3)

- We showcase with numerical experiments on pretrained models and real-world data that BaLoRA outperforms traditional Low-Rank Adaptation and is more robust to variations in the learning rate or the scaling at initialization. (Section 4)

## 1.1 RELATED WORK

**Parameter-efficient fine-tuning.** To enable efficient adaptation without full retraining, residual adapters were proposed for computer vision tasks (Rebuffi et al. (2017)) and later extended to

NLP through adapter-based transfer learning (Houlsby et al., 2019). Other PEFT strategies include pruning-based approaches (Diff-Pruning, Guo et al. (2020)) and low-rank adapters (Hu et al., 2022). LoRA and its variants have been widely applied, from bridging language models with non-language tasks through LIFT (Dinh et al., 2022), to fine-tuning image generation models (Fan et al., 2023). Theoretical analyses of LoRA in the NTK regime have been conducted by Malladi et al. (2023); Jang et al. (2024), and its expressive power has been studied by Zeng & Lee (2024).

**LoRA optimization.** LoRA is typically trained with Adam (Kingma & Ba (2017)) or AdamW (Loshchilov & Hutter (2017)). Several recent works have proposed to tailor the optimization problem to low-rank structures. Riemannian approaches (Bogachev et al., 2025; Mo et al., 2025) address overparameterization through manifold-based optimization, but often require specialized algorithms. Other studies on matrix factorization have investigated alternative algorithms to improve convergence guarantees: Zhang & Fan (2024) analyze projected gradient descent, showing convergence rates independent of condition numbers under certain assumptions; Ward & Kolda (2023) study alternating gradient descent, with bounds depending on spectral gaps; Zhang & Pilanci (2024) modify gradient updates by incorporating preconditioners derived from Riemannian optimization; and Olikier et al. (2025) propose Gauss–Southwell type descent methods, focusing on the interaction between step-size rules and balancing. While matrix factorization can be viewed as a special case of LoRA, these studies do not address fine-tuning of machine learning models.

**Initialization and convergence dynamics.** Standard LoRA initialization sets one low-rank matrix to zero and the other to Gaussian noise. This ensures the model initially behaves like the pre-trained model, while allowing low-rank adaptations during training. The initial update $A_0 B_0$ is then scaled by a factor $\alpha/r$, where $\alpha$ is a hyperparameter (Hu et al., 2022). Rank-stabilized scaling can be implemented to prevent gradient collapse at higher ranks (Kalajdzievski, 2023). The convergence of LoRA is closely related to results on deep linear networks and matrix factorization. For small step sizes, gradient descent converge under balancing conditions (Nguegnang et al., 2024), which become exact conservation laws of the gradient flow in the vanishing step size limit, explaining implicit biases (Marcotte et al., 2023). Random initialization can ensure global convergence in asymmetric low-rank matrix factorization (Ye & Du, 2021). Large step sizes can push training toward the edge of stability (Cohen et al., 2021), a phenomenon extensively analyzed in linear networks (Ghosh et al., 2025; Chen & Bruna, 2023). For LoRA specifically, Hayou et al. (2024b) assign different learning rates to the low-rank factors for more efficiency, and Xu et al. (2025) analyze its dynamics in matrix factorization with a gradient flow perspective, showing an initial alignment phase followed by a local convergence phase for small initialization scales.

**Structural constraints.** Low-rank-based models are inherently overparameterized, and the resulting inefficiencies can be mitigated by adding structural constraints. Orthogonality has been considered for optimization on the Stiefel manifold (Park et al., 2025) and QR-based initialization (OLoRA, Büyükakyüz (2024)). Other approaches leverage richer decompositions: DoRA (Liu et al. (2024)) decomposes weights into magnitude and direction, butterfly-based orthogonal fine-tuning (BOFT) (Liu et al. (2023b)) and Householder reflection adaptation (HRA) (Yuan et al. (2024)) leverage structured orthogonal parameterizations, SVFT (Lingam et al. (2024)) exploits singular vectors of pre-trained weights, and VeRA (Kopiczko et al. (2023)) reduces parameters by sharing low-rank random matrices with compact scaling vectors. GOAT (Fan et al. (2025)) uses SVD-structured priors with mixture-of-experts alignment to improve initialization and scaling, while LoRA Done RITE (Yen et al. (2024)) enforces invariance of the optimization process under scaling and rotation transformations of adapters.

## 2 BALANCED MINIMIZERS ARE BEST CONDITIONED

In this section, we analyze the asymptotic convergence rate of LoRA in a tractable setting. We consider a pre-trained one-layer linear network $W \in \mathbb{R}^{n \times m}$ and a *target* $W^\star \in \mathbb{R}^{n \times m}$, representing the ideal fine-tuned model (Zeng & Lee, 2024). Let $Z := W^\star - W$ the gap between the pretrained and target models. LoRA then consist in minimizing the loss

$$f \colon (A, B) \in \mathbb{R}^{n \times r} \times \mathbb{R}^{r \times m} \mapsto \frac{1}{2}\|Z - AB\|_F^2 \,, \tag{1}$$

with $\|\cdot\|_F$ denoting the Frobenius norm. To solve (1), one can use gradient descent (GD): with step size $\gamma$ and initialization $(A_0, B_0)$, the updates read

$$\begin{cases} A_{t+1} = A_t - \gamma \nabla_{A_t} f(A_t, B_t) = A_t - \gamma(A_t B_t - Z)B_t^\top, \\ B_{t+1} = B_t - \gamma \nabla_{B_t} f(A_t, B_t) = B_t - \gamma A_t^\top(A_t B_t - Z). \end{cases} \quad (2)$$

This setup generalizes matrix factorization (Ye & Du, 2021; Ghosh et al., 2025), but unlike that setting (where $\mathrm{rk}\, Z = r$), $Z$ may have rank $\mathrm{rk}\, Z \geq r$. This introduces off-diagonal terms in the Hessian and makes the mathematical analysis more involved.

The iterations (2) are known to converge to a minimizer $(A^\star, B^\star)$ of $f$ (Nguegnang et al., 2024), but due to LoRA's overparameterization, several minimizers correspond to the same optimal adapter $(AB)^\star$. These minimizers then form the $r^2$-dimensional manifold $\mathcal{S}_Z = \{(A, B) \in \mathbb{R}^{n \times r} \times \mathbb{R}^{r \times m} : AB = LR_r(Z)\}$, where $LR_r(Z) := \sum_{i=1}^r \sigma_i u_i v_i^\top$ is the best rank-$r$ approximation of $Z$ in Frobenius norm, obtained from the singular value decomposition (SVD) of $Z$. To identify which minimizers allow the fastest convergence, we study the *condition number* of $f$, *i.e.*, $\kappa(f)(A^\star, B^\star) := L/\mu$, where, denoting $H(A^\star, B^\star) := D^2 f(A^\star, B^\star)$ the Hessian of $f$ at a minimizer $(A^\star, B^\star)$, $L := \lambda_{\max}(H(A^\star, B^\star))$ is its largest eigenvalue, and $\mu := \lambda_{\min \neq 0}(H(A^\star, B^\star))$ its smallest non-zero eigenvalue. The following classical result shows that a smaller $\kappa(f)(A^\star, B^\star)$ implies faster asymptotic convergence (*e.g.*, Bach (2024)).

**Lemma 2.1.** *Consider the iterations (2) and let $(A^\star, B^\star)$ denote their limit, which is a minimizer of $f$ (Nguegnang et al., 2024). Define $L := \lambda_{\max}(H(A^\star, B^\star))$ and $\mu := \lambda_{\min \neq 0}(H(A^\star, B^\star))$. Then,*

$$\limsup_{t \to +\infty} \frac{f(A_{t+1}, B_{t+1}) - f(A^\star, B^\star)}{f(A_t, B_t) - f(A^\star, B^\star)} \leq \max((1 - \gamma\mu)^2, (1 - \gamma L)^2).$$

*Taking $\gamma = 2/(L + \mu)$ to minimize the right-hand side, and denoting $\kappa := \kappa(f)(A^\star, B^\star) = L/\mu$,*

$$\limsup_{t \to +\infty} \frac{f(A_{t+1}, B_{t+1}) - f(A^\star, B^\star)}{f(A_t, B_t) - f(A^\star, B^\star)} \leq \left(\frac{\kappa - 1}{\kappa + 1}\right)^2.$$

*Hence, the smaller $\kappa \geq 1$, the faster the convergence to $(A^\star, B^\star)$ asymptotically.*

Among all minimizers in $\mathcal{S}_Z$, we single out the submanifold of *balanced minimizers*:

$$\mathcal{B}_Z := \mathcal{B} \cap \mathcal{M}_Z \quad \text{where} \quad \mathcal{B} := \{(A, B) \in \mathbb{R}^{n \times r} \times \mathbb{R}^{r \times m} : A^\top A = BB^\top\}. \quad (3)$$

As we show next, these balanced minimizers achieve an optimal condition number and therefore lead to the fastest local convergence.

## 2.1 THE MATRIX FACTORIZATION CASE

When $\mathrm{rk}\, Z = r$, the problem reduces to matrix factorization, and the minimal value of the loss $f$ is zero. We explicitly compute the Hessian of $f$ at any minimizer $(A, B)$ (we remove the stars to lighten the notation), and determine its full spectrum and corresponding condition number.

**Proposition 2.1.** *Let $(A, B) \in \mathbb{R}^{n \times r} \times \mathbb{R}^{r \times m}$ be a global minimizer of the loss $f$ (1). Assume $\mathrm{rk}\, Z = r$. The Hessian of $f$ at $(A, B)$ reads*

$$H(A, B) = \begin{pmatrix} (BB^\top) \otimes I_n & B \otimes A \\ B^\top \otimes A^\top & I_m \otimes (A^\top A) \end{pmatrix},$$

*and its spectrum consists of 0 (with multiplicity $r^2$), $\sigma_i(A)^2 + \sigma_j(B)^2$ for $1 \leq i, j \leq r$ (each with multiplicity 1), $\sigma_i(A)^2$ (each with multiplicity $m - r$), and $\sigma_j(B)^2$ (each with multiplicity $n - r$). Therefore, $\kappa(f)(A, B) = (\sigma_1(A)^2 + \sigma_1(B)^2)/\min(\sigma_r(A)^2, \sigma_r(B)^2)$.*

Proposition 2.1 establishes a direct link between the condition number of a minimizer $(A, B)$ and its degree of balancedness, which allows us to identify optimally conditioned minimizers.

**Proposition 2.2.** *Assume $\mathrm{rk}\, Z = r$. The minimizers of $f$ with minimal condition number belong to $\{(A, B) : AB = LR_r(Z), \sigma_i(A) = \sigma_i(B), u_i^B = v_i^A \text{ for } i \in \{1, r\}\}$, where $(u_i^B)$ are the left singular vectors of $B$ and $(v_i^A)$ the right singular vectors of $A$. In particular, all balanced minimizers (3) have minimal condition number $\kappa_{\min} = 2\sigma_1(Z)/\sigma_r(Z)$.*

Combining Proposition 2.2 with Lemma 2.1 yields a closed-form connection between the best asymptotic convergence rate of the dynamics (2) and the spectrum of the target matrix $Z$. In particular, it shows that a target with a more spread-out spectrum corresponds to a more challenging matrix factorization problem. Furthermore, our analysis identifies balanced minimizers as those achieving optimal (*i.e.*, minimal) conditioning, highlighting their key role in the convergence behavior of the dynamics. To the best of our knowledge, these explicit quantitative links between spectral structure, conditioning, and convergence rates have not been reported previously.

## 2.2 THE GENERAL CASE

We now investigate how our insights for matrix factorization extend to the general case $\mathrm{rk}\, Z \geq r$, where $Z$ cannot be exactly represented with rank-$r$. Here, the residual $AB - Z \neq 0$ introduces additional off-diagonal terms in the Hessian, making its diagonalization more involved.

**Proposition 2.3.** *Let* $(A, B) \in \mathbb{R}^{n \times r} \times \mathbb{R}^{r \times m}$ *be a global minimizer of the loss $f$ (1). Assume* $\mathrm{rk}\, Z \geq r$. *The Hessian of $f$ at $(A, B)$ reads*

$$H(A, B) = \begin{pmatrix} (BB^\top) \otimes I_n & B \otimes A + (I_r \otimes (AB - Z))K_{r,m} \\ B^\top \otimes A^\top + ((AB - Z)^\top \otimes I_r)K_{n,r} & I_m \otimes (A^\top A) \end{pmatrix}$$

*where $K_{a,b}$ is the $ab \times ab$ matrix such that $\mathrm{vec}(X^\top) = K_{a,b}\mathrm{vec}(X)$ for any $X \in \mathbb{R}^{a \times b}$, with $\mathrm{vec}$ the vectorization operator.*

One can verify that $H(A, B)$ is symmetric, since $(I_r \otimes (AB - Z))K_{r,m}(x \otimes y) = (I_r \otimes (AB - Z))(y \otimes x) = y \otimes (AB - Z)x = K_{n,r}^\top((AB - Z)x \otimes y)$, as $K_{n,r}^\top = K_{r,n}$. The off-diagonal terms now couple the $A$ and $B$ blocks through the residual, which is a key difference from the matrix factorization case. As a result, characterizing the full spectrum of $H(A, B)$ is more challenging. Below, we compute the sharpness of the Hessian at a minimizer and provide two bounds on its smallest eigenvalue. The proof is detailed in Appendix A.2.

**Proposition 2.4.** *Let* $(A, B) \in \mathbb{R}^{n \times r} \times \mathbb{R}^{r \times m}$ *be a global minimizer of the loss $f$ (1). The largest eigenvalue of the Hessian of $f$ at $(A, B)$ is $\lambda_{\max}(H(A, B)) = \sigma_1(A)^2 + \sigma_1(B)^2$. Moreover, the smallest non-zero eigenvalue of $H(A, B)$ satisfies,*

$$\min(\sigma_r(A)^2, \sigma_r(B)^2) - \sigma_{r+1}(Z) \leq \lambda_{\min \neq 0}(H(A, B)) \leq \min(\sigma_r(A)^2, \sigma_r(B)^2). \tag{4}$$

*If $(A, B)$ is balanced, the lower bound in (4) is* maximized*, equal to $\sigma_r(Z) - \sigma_{r+1}(Z)$, and becomes an* equality*: $\lambda_{\min \neq 0}(H(A, B)) = \sigma_r(Z) - \sigma_{r+1}(Z)$.*

Compared to matrix factorization, balanced minimizers are still optimally conditioned, but the key quantity governing the intrinsic hardness of LoRA optimization shifts from $\sigma_r(Z)$ to the $r$-spectral gap $\sigma_r(Z) - \sigma_{r+1}(Z)$. This gap quantifies how well the rank-$r$ approximation separates from the discarded directions. The smaller the gap (and the larger $\sigma_1(Z)$), the slower the asymptotic convergence of the iterations (2) in the best case.

Our results in this section suggest that driving the dynamics towards balanced adapters (2), through explicit or implicit regularization, can accelerate training in practice. Next, we leverage this insight to design a fine-tuning strategy that guides LoRA along balanced adapters, resulting in faster convergence and improved stability in practice.

## 3 BALORA: BALANCED LOW-RANK ADAPTATION

Section 2 highlights that balanced minimizers $(A, B) \in \mathcal{B}$ achieve optimal condition number, where $\mathcal{B}$ is the *balanced manifold* (Du et al., 2018) defined in Equation (3). We thus propose to constrain LoRA iterations to stay on $\mathcal{B}$ by projecting the iterates after each gradient (or Adam) step, to ensure convergence to a better-conditioned minimizer and faster asymptotic rates. As exposed in the following section, we introduce a submanifold $\mathcal{H} \subset \mathcal{B}$ of *hyperbalanced* matrices, which provides a more structured and efficient parameterization. We call Balanced Low-Rank Adaptation (BaLoRA) the novel fine-tuning method obtained by projecting to $\mathcal{H}$. In the remainder, we will use the term "manifold" with slight abuse of language, as the sets considered are in fact manifolds with boundary.

---

**Algorithm 1** Balanced projection

---

**Require:** $(A, B) \in \mathbb{R}^{n \times r} \times \mathbb{R}^{r \times n}$
 1: Compute polar decompositions $A = R_A S_A$ and $B = S_B R_B$
 2: Compute $S = S_A S_B \in \mathbb{R}^{r \times r}$
 3: Compute SVD decomposition $S = U \Sigma V^\top$
 4: **return** matrices $A^{\mathrm{proj}}, B^{\mathrm{proj}} = R_A(U \Sigma^{1/2}), (\Sigma^{1/2} V^\top) R_B$

---

### 3.1 Hyperbalanced Manifold and the BaLoRA Mapping

Let $\mathbb{D}_+^r$ denote the set of $r \times r$ non-negative diagonal matrices with ordered diagonal values. We consider the submanifold (with boundary) $\mathcal{H} \subset \mathcal{B}$, which we call the *hyperbalanced manifold*

$$\mathcal{H} := \{(A, B) \in \mathbb{R}^{n \times r} \times \mathbb{R}^{r \times m} : \exists S \in \mathbb{D}_+^r \text{ s.t. } A^\top A = BB^\top = S\}.$$

As detailed in Proposition B.1 proved in the appendices, one has the equivalent description of $\mathcal{H}$:

$$\mathcal{H} = \{(US^{1/2}, S^{1/2}V) : U^\top U = VV^\top = I_r, \; S \in \mathbb{D}_+^r\}. \tag{5}$$

This reformulation has two major consequences.

**Consequence 1: optimizing on $\mathcal{H}$ is equivalent to optimizing over low-rank matrices.** Denoting $\mathcal{N}_r$ the set of rank-$r$ matrices, (5) shows that $(A, B) \in \mathcal{H} \mapsto X := AB \in \mathcal{N}_r$ is surjective. For a function $g(X)$, define $f(A, B) := g(AB)$. Then, $\min_{X \in \mathcal{N}_r} g(X)$ and $\min_{(A,B) \in \mathcal{H}} f(A, B)$ are equivalent problems. Working with variables $(A, B) \in \mathcal{H}$ therefore provides a computationally convenient framework for low-rank optimization, while also taking advantage of the improved conditioning discussed in the previous section.

**Consequence 2: definition of the BaLoRA map $P$ "projecting" onto $\mathcal{H}$.** Given $(A, B)$ with $X := AB$, take any reduced SVD with strictly ordered singular values $X = U S V^\top$. The *BaLoRA map* is defined as

$$P(A, B) := (US^{1/2}, S^{1/2}V^\top). \tag{6}$$

The reformulation in (5) shows that $P$ "projects" onto $\mathcal{H}$. Although this is not an orthogonal projector, Proposition B.2 in the appendix shows that it exhibits a "projection-like" behavior, namely, it defines a smooth retraction (Absil & Malick, 2012) onto $\mathcal{H}$. Consequently, results from Riemannian optimization (Boumal, 2023) can be applied to analyze the convergence of the BaLoRA-GD method, *i.e.*, gradient descent combined with $P$ to keep the iterates on $\mathcal{H}$, as detailed in the next section. Another important property of the BaLoRA map $P$ is that it preserves the product, unlike the orthogonal projector: denoting $(\tilde{A}, \tilde{B}) := P(A, B)$, then $\tilde{A}\tilde{B} = AB$. This preservation guarantees that the loss remains *unchanged*, which is essential for the intrinsic reformulation described in Section 3.2. The procedure to efficiently compute $P$ is given in Algorithm 1, and has computational complexity $\mathcal{O}(nr^2)$ for $n \gg r$, which adds negligible overhead to the cost of the optimizer step (Figure 4).

### 3.2 The BaLoRA method and BaLoRA-GD

> **Definition.** The *BaLoRA* method consists in applying the map
> $P(A, B) = (US^{1/2}, S^{1/2}V^\top)$ at the end of each step of an optimization scheme, where
> $AB = USV^\top$. When combined with Adam, we simply refer to the resulting algorithm
> as *BaLoRA*. When applied to the iterates of gradient descent, we call it *BaLoRA-GD*.

While BaLoRA (with Adam) is a heuristic method whose theoretical analysis is beyond the scope of this work, we show that the gradient descent variant, BaLoRA-GD, exhibits a striking intrinsic behavior. Recall that BaLoRA-GD iterates for $k \geq 0$ and some stepsize $\tau_k > 0$ as $(A_{k+1}, B_{k+1}) := P(A_k - \tau_k \nabla_A f(A_k, B_k), B_k - \tau_k \nabla_B f(A_k, B_k))$, starting from any initialization $(A_0, B_0)$.

**Intrinsic BaLoRA-GD.** Consider a loss function of the form $f(A, B) = g(X)$, with $X = AB$ and $g: \mathbb{R}^{n \times m} \to \mathbb{R}$ smooth. This general setting encompasses all LoRA losses. Then, the BaLoRA-GD iteration can be expressed entirely as an intrinsic algorithm on the low-rank manifold $\mathcal{N}_r$. Interestingly, it is not a pure gradient descent, since the update includes a $\tau_k^2$ correction term. This correction

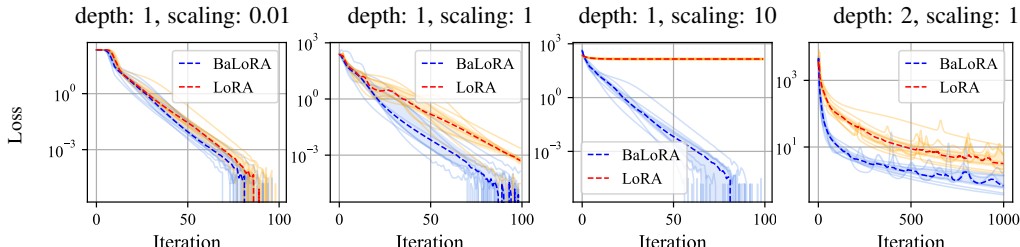

Figure 2: **Matrix factorization.** Evolution of the loss of LoRA with GD vs. GD-BaLoRA in different settings. The dotted lines are the median of 8 curves for each method, each curve corresponding to a different seed for the initialization and the optimal learning rate for this seed, while the target is kept fixed. Both methods use the traditional LoRA init, with $A_0 = 0$, $B_0$ random Gaussian, and a scaling $\alpha/r$. The first three plots corresponds to a square one-layer linear network of size 20, with a lora rank of 4, and $\alpha \in \{0.01r, r, 10r\}$. The fourth plot was obtained with a two-layer linear network of size 20, a LoRA rank of 4, and $\alpha = r$. BaLoRA converges faster in all situations, with a particular advantage when the scaling $\alpha$ is large (third plot), where LoRA fails to converge.

is essential for the method to be implementable in terms of the factor variables $(A, B)$. The connection with $\mathcal{N}_r$ is valuable for understanding the behavior of the method, but in practice one should rely on the original BaLoRA-GD formulation, particularly because it can be directly adapted to its Adam variant, intended for large-scale LoRA adaptation of transformers.

**Proposition 3.1** (Intrinsic update on $X = AB$). *For $f(A, B) = g(AB)$, denoting $(A_k, B_k)$ the BaLoRA-GD iteration, then for $k \geq 1$, $X_k := A_k B_k$ satisfies*

$$X_{k+1} = X_k - \tau_k\Big((X_k X_k^\top)^{1/2}\, G_k + G_k\, (X_k^\top X_k)^{1/2}\Big) + \tau_k^2\, G_k X_k^\top G_k, \qquad G_k = \nabla g(X_k). \quad (7)$$

*Note that (7) might not hold for $k = 0$, since $(A_0, B_0)$ might not belong to $\mathcal{H}$.*

*Proof.* Since $f(A, B) = g(AB)$, the chain rule gives $\nabla_A f(A_k, B_k) = G_k B_k^\top$, $\nabla_B f(A_k, B_k) = A_k^\top G_k$. Hence the pre-projection product is

$$\widetilde{X}_k := \widetilde{A}_k \widetilde{B}_k = X_k - \tau_k\big(A_k A_k^\top G_k + G_k B_k^\top B_k\big) + \tau_k^2\, G_k X_k^\top G_k.$$

By construction of $P$, the product is preserved by $P$, so $X_{k+1} = A_{k+1} B_{k+1} = \widetilde{X}_k$. Because $(A_k, B_k) \in \mathcal{H}$, write an SVD $X_k = U_k S_k V_k^\top$ and balanced factors $A_k = U_k S_k^{1/2}$, $B_k = S_k^{1/2} V_k^\top$. Then $A_k A_k^\top = U_k S_k U_k^\top = (X_k X_k^\top)^{1/2}$, $B_k^\top B_k = V_k S_k V_k^\top = (X_k^\top X_k)^{1/2}$. Substitute these into $\widetilde{X}_k$ to obtain Equation (7). $\qquad\square$

**Riemannian flow.** To gain geometric insight into BaLoRA-GD, we look at the small–stepsize limit. Set $\tau_k \equiv \tau$ and $t = k\tau$, taking $\tau \to 0$ in Equation (7) yields the continuous-time dynamics

$$\dot{X} = -H_X[\nabla g(X)], \qquad H_X[W] := (XX^\top)^{1/2}W + W(X^\top X)^{1/2}$$

on the rank-$r$ manifold $\mathcal{N}_r$. This ODE is the gradient flow of $g$ with respect to the Riemannian metric $\langle U, V \rangle_X = \langle U, H_X^{-1}[V] \rangle_F$ (here $H_X$ is self-adjoint and positive on the tangent space). When restricted to symmetric positive definite (SPD) matrices, $H_X[W] := XW + WX$ is the inverse of the Bures metric. This metric is well known in optimal transport on Gaussian measures and provides a natural tool for optimization over the cone of SPD matrices (Bhatia et al., 2019).

## 4 EXPERIMENTS

We present an empirical evaluation of BaLoRA, demonstrating its benefits in terms of performance and convergence speed, with negligible computational overhead. Our experiments cover fine-tuning tasks on both synthetic and real-world data, with different pre-trained architectures. We also provide ablation studies that highlight the robustness of BaLoRA with respect to hyperparameter choice.

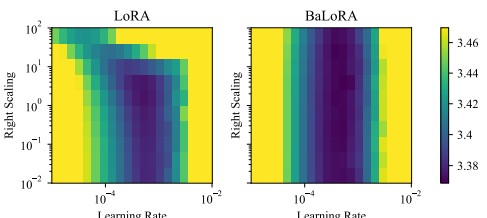 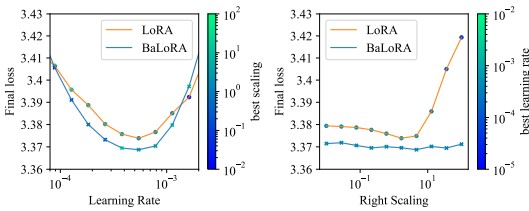

(a) Final test loss vs. (learning rate, scaling)  (b) Best test loss per learning rate or scaling

Figure 3: **LLM finetuning results.** Final test loss of LoRA and BaLoRA for GPT-2 finetuning on Wikitext-2-raw-v1. In (3b), the marker color indicates the magnitude of the optimal initialization scaling (3b, left) or learning rate (3b, right): darker shades correspond to smaller values. While the optimal learning rate is approximately constant across scalings for both methods, BaLoRA systematically selects larger initialization scalings compared to LoRA.

## 4.1 SYNTHETIC EXPERIMENTS

We first compare the dynamics of GD-BaLoRA and standard (GD) LoRA on a toy framework which aligns closely with our theoretical analysis. Specifically, we consider the optimization problems,

$$\min_{(A,B)} \|W^\star - (W + AB)\|_F^2 \qquad \text{and} \qquad \min_{(A,B)} \|W^\star - (W_1 + A_1 B_1)(W_2 + A_2 B_2)\|_F^2$$

for $(A, B) \in \mathbb{R}^{n \times r} \times \mathbb{R}^{r \times n}$. The first problem corresponds to the setting studied in Section 2, while the second extends it to a two-layer linear network and introduces more complex interactions between layers. We apply LoRA and BaLoRA to these problems across several initialization seeds. For each seed and each method, we train over a grid of learning rates and scaling values $\alpha$. In Figure 2, we report the training loss over iterations for the learning rate that achieves the lowest loss. We see that for both configurations (one or two layers), BaLoRA outperforms LoRA. This confirms our insights from Section 2 and extends their scope to the two-layer case. Moreover, the performance gap grows for larger initialization scaling $\alpha$, where LoRA fails to converge while BaLoRA remains stable. For small scaling values ($\sim 10^{-2}$), however, LoRA and BaLoRA exhibit very similar convergence rates, a phenomenon that our current analysis does not explain. Note that the conclusions above also hold for a fixed learning rate, as shown in Figure 6.

## 4.2 EXPERIMENTS WITH LARGE LANGUAGE MODELS

Next, we scale up the experiments to large language models and real-world data. We track the train and test loss of LoRA versus BaLoRA during the fine-tuning, as a function of the runtime. We believe that reporting loss values is the right way to evaluate the efficiency of BaLoRA as a fine-tuning algorithm.

**Setup.** We fine-tune the pre-trained Huggingface models GPT-2 (Radford et al., 2019) and Llama-3.2-3B (Meta AI, 2024) using LoRA and BaLoRA, to perform next-token prediction on the Wikitext-2-raw-v1 dataset (Merity et al., 2016). The training split contains 36.7k samples, and we set a context length of 1024 tokens. Test loss is measured as cross-entropy on a 1000-sample subset of the test split. We simultaneously fine-tune all MLP layers with AdamW (Loshchilov & Hutter, 2017) at a constant learning rate, while keeping the attention layers frozen. We use a constant LoRA rank of 8. The training batch size is 16 for GPT-2, with one gradient accumulation step, and 8 for Llama-3.2, with 4 gradient accumulation steps. For each model, we perform an ablation study over a two-dimensional grid of learning rates and initialization scalings. This allows us to analyze the impact of these hyperparameters on convergence, and demonstrates that BaLoRA is more robust to hyperparameter variations, in particular to changes in scaling.

**Results and discussion.** We provide three different types of plots to analyze the behavior of BaLoRA compared to LoRA. Figure 3 (resp. Figure 7) reports the final test loss after fine-tuning GPT-2 (resp. Llama-3.2) on Wikitext with 2 epochs. The left panels sweep jointly over learning rates and initialization scalings, while the right panels marginalize one variable to show the best achievable loss for each fixed learning rate (resp. scaling). From these plots, we see that BaLoRA consistently achieves lower test loss than LoRA when each method is tuned over both hyperparameters.

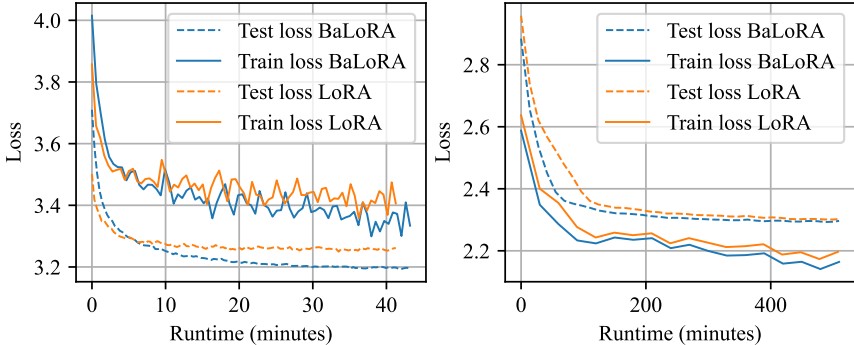

Figure 4: **Training curves vs. time.** BaLoRA outperforms LoRA, sometimes by a large margin, despite the small computational overhead. **Left:** 5 epochs on GPT-2, with a learning rate of $3.8 \times 10^{-4}$ and an init scaling of $35.9$. The large scaling significantly advantages BaLoRA. **Right:** 2 epochs on Llama-3.2, with the best choice of hyperparameters for each method. For LoRA, the learning rate is $2.2 \times 10^{-5}$ and the scaling is $0.01$. For BaLoRA, the learning rate is $4.6 \times 10^{-5}$ and the scaling is $1$.

|  | GPT-2 | | | Llama-3.2-3B |
|---|---|---|---|---|
|  | epoch 1 | epoch 2 | epoch 5 | epoch 1 |
| LoRA | 3.387 | 3.374 | 3.370 | 2.302 |
| BaLoRA | **3.382** | **3.368** | **3.367** | **2.296** |

Table 1: Final test loss of LoRA and BaLoRA when finetuning GPT-2 and Llama-3.2-3B on Wikitext-2-raw-v1. BaLoRA consistently outperforms LoRA.

Moreover, BaLoRA displays enhanced robustness: while LoRA deteriorates rapidly under large initialization scalings, BaLoRA remains stable and achieves nearly constant performance across a wide scaling range. This robustness extends, though more mildly, to variations in the learning rate. These observations suggest that BaLoRA alleviates part of the delicate hyperparameter tuning burden, and point a distinctive and potentially useful property to explore in future work. Figure 4 focuses on training loss and test loss trajectories as a function of the runtime. In the left plot, obtained with GPT-2, we take the same learning rate and initialization scaling for LoRA and BaLoRA, choosing a large scaling, which advantages BaLoRA. We observe a significant difference in the final test loss, as well as in the convergence rates of the train losses, despite the small overhead induced by the BaLoRA projection. The right plot, obtained with Llama-3.2, plots train and test curves for the best choice of scaling and learning rate for each method. The difference between LoRA and BaLoRA is therefore smaller in the test loss, while BaLoRA's train loss remains significantly smaller than LoRA's. We also report the final test loss value corresponding to the best choice of hyperparameters for each model and method, in Table 1.

## 5 CONCLUSION

In this paper, we conducted a theoretical analysis on the LoRA convergence dynamics. We showed that its overparameterization produces minimizers with varying condition numbers, and that balanced minimizers achieve optimal conditioning. Building on this insight, we proposed BaLoRA, an extension of LoRA that enforces balance by projecting adapters onto the hyperbalanced manifold after each optimization step. This projection preserves the adapted weight matrix while improving conditioning, yielding faster convergence and greater robustness to hyperparameter choices with negligible overhead. Empirical results on large language models (GPT-2, Llama-3.2) confirm that BaLoRA consistently outperforms LoRA, both in terms of final accuracy and stability across learning rates and initialization scalings. These findings highlight balance as a key structural property in low-rank adaptation and demonstrate that explicitly enforcing it leads to practical gains. On the theoretical side, extending our analysis to multi-layer and nonlinear settings could provide deeper insight into how balance interacts with modern architectures. From an optimization perspective, BaLoRA connects naturally to geometry-aware methods and may inspire new algorithms on matrix manifolds. Finally, investigating whether enforcing balance impacts generalization or robustness to distribution shift could broaden its relevance beyond convergence speed.

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

## A  POSTPONED PROOFS

### A.1  PROOF OF PROPOSITION 2.1

Computing $H(A, B)$ is straightforward. To diagonalize this matrix, notice that $H(A, B) = MM^\top$ with

$$M := \begin{pmatrix} B \otimes I_n \\ I_m \otimes A^\top \end{pmatrix} \in \mathbb{R}^{r(m+n) \times mn}.$$

- **Kernel of** $H(A, B)$**.** The kernel of $H(A, B)$ is equal to the kernel of $M^\top$, which can be written as $\{\text{vect}\,(AR, -RB) : R \in \mathbb{R}^{r \times r}\}$. Indeed, unvectorizing the equation $M^\top \text{vect}\,(D, E) = 0$ gives $DB + AE = 0$, and such $(D, E)$ can be rewritten as $(AR, -RB)$ for $R = A^+D = -EB^+$. Therefore, $\ker H(A, B)$ is of dimension $r^2$.

- **Non-zero spectrum of** $H(A, B)$**.** To find the non-zero eigenvalues of $H(A, B)$ and associated eigenvectors, we use the following observation.

  **Lemma A.1.** *Let $M$ be a matrix and $x$ be an eigenvector of $M^\top M$ associated with a non-zero eigenvalue $\lambda$. Then $Mx$ is an eigenvector of $MM^\top$, associated with the eigenvalue $\lambda$.*

  We have
  $$M^\top M = (B^\top B) \otimes I_n + I_m \otimes (AA^\top).$$

  Let $(\lambda, x)$ be an eigenpair of $AA^\top$ and $(\mu, y)$ an eigenpair of $B^\top B$. Then $(\lambda + \mu, y \otimes x)$ is an eigenpair of $M^\top M$. Therefore, denoting $\lambda_1, \ldots, \lambda_r$ and $\mu_1, \ldots, \mu_r$ the non-zero eigenvalues of $AA^\top$ and $B^\top B$ respectively, with associated unit eigenvectors $x_1, \ldots, x_r$ and $y_1, \ldots, y_r$, and denoting $x_{r+1}, \ldots, x_n$ and $y_{r+1}, \ldots, y_m$ unit bases of $\ker AA^\top$ and $\ker B^\top B$, the eigenpairs of $M^\top M$ associated with non-zero eigenvalues are

  $$(\lambda_i + \mu_j, y_j \otimes x_i)_{1 \le i,j \le r} \cup (\lambda_i, y_{r+j} \otimes x_i)_{\substack{1 \le j \le m-r \\ 1 \le i \le r}} \cup (\mu_j, x_j \otimes x_{r+i})_{\substack{1 \le i \le n-r \\ 1 \le j \le r}}.$$

  Using Lemma A.1, the eigenpairs of $MM^\top$ with non-zero eigenvalues are thus

  $$(\lambda_i + \mu_j, M(y_j \otimes x_i))_{1 \le i,j \le r} \cup (\lambda_i, M(y_{r+j} \otimes x_i))_{\substack{1 \le j \le m-r \\ 1 \le i \le r}} \cup (\mu_j, M(y_j \otimes x_{r+i}))_{\substack{1 \le i \le n-r \\ 1 \le j \le r}}.$$

## A.2 PROOF OF PROPOSITION 2.4

**Sharpness of the Hessian.** Let us first compute the largest eigenvalue of $H(A, B)$. Denote $H_1 :=$ $\begin{pmatrix} (BB^\top)\otimes I_n & B\otimes A \\ B^\top\otimes A^\top & I_m\otimes(A^\top A) \end{pmatrix}$ and $H_2 := \begin{pmatrix} 0_{nr\times nr} & (I_r\otimes(AB-Z))K_{r,m} \\ ((AB-Z)^\top\otimes I_r)K_{n,r} & 0_{mr\times mr.} \end{pmatrix}$, so that $H(A, B) = H_1 + H_2$.

**Lemma A.2.** *The largest eigenvalue of $H_2$ is $\sigma_{r+1}(Z)$. The smallest eigenvalue of $H_2$ is $-\sigma_{r+1}(Z)$.*

*Proof.* Denote $G := (I_r \otimes (AB - Z))K_{r,m} = (((AB - Z)^\top \otimes I_r)K_{n,r})^\top \in \mathbb{R}^{rn\times rm}$. Let $(u_1, \ldots, u_{rn})$ and $(v_1, \ldots, v_{rm})$ be respectively the left and right eigenvectors of $G$. Denote $\tau_1 \geq \cdots \geq \tau_\varrho$ the singular values of $G$, with $\varrho := \mathrm{rk}\, G = r(\min(n, m) - r)$.

- The Kernel of $H_2$ is the span of $\{\binom{u_j}{0}: j = \varrho+1, \ldots, nr\} \cup \{\binom{0}{v_k}: k = \varrho+1, \ldots, mr\}$.

- For $i = 1, \ldots, \varrho$, let $x_i^+ := \binom{u_i}{v_i}$ and $x_i^- := \binom{u_i}{-v_i}$. Then $x_i^+$ (resp. $x_i^-$) is an eigenvector of $H_2$ associated with the eigenvalue $\sigma_i$ (resp. $-\sigma_i$). The $\sigma_i$ can be easily computed, they are of the form $-\sigma_k(Z)$ for $k = r + 1, \ldots, \min(n, m)$, which proves the result.

$\square$

The eigenvectors of $H_1$ form a basis of the space $\mathbb{R}^{(n+m)r}$. We will prove that for any eigenvector $u$ of this basis, it holds $|Hu| \leq (\sigma_1(A)^2 + \sigma_1(B)^2)|u|$.

- If $u$ is in the kernel of $H_1$ and has unit norm, then $|Hu| = |H_2u| \leq \|H_2\|_2 = \sigma_{r+1}(Z) \leq \sigma_1(Z) \leq \sigma_1(A)\sigma_1(B) \leq \sigma_1(A)^2 + \sigma_1(B)^2$.

- If $u$ is of the form $M(y_j \otimes x_i)$ with the notations of the proof of Proposition 2.1, then

$$H_2u = \begin{pmatrix} (I_r \otimes (AB - Z))(A^\top x_i \otimes y_j) \\ ((AB - Z)^\top \otimes I_r)(x_i \otimes By_j) \end{pmatrix} = 0, \tag{8}$$

so $|Hu| = |H_1u| \leq (\sigma_1(A)^2 + \sigma_1(B)^2)|u|$.

- If $u$ is of the form $M(y_{r+j} \otimes x_i)$ for $1 \leq i \leq r$ and $1 \leq j \leq m - r$, it is easy to check that $H_1u$ and $H_2u$ are orthogonal. Then:

$$\begin{aligned} |Hu| &= \sqrt{|H_1u|^2 + |H_2u|^2} \\ &\leq \sqrt{\sigma_i(A)^4|u|^2 + \sigma_{r+1}(Z)^2|u|^2} \\ &= \sqrt{\sigma_i(A)^4 + \sigma_{r+1}(Z)^2}|u|. \end{aligned}$$

We therefore need to prove that $\sqrt{\sigma_i(A)^4 + \sigma_{r+1}(Z)^2} \leq \sigma_1(A)^2 + \sigma_1(B)^2$, or equivalently that $\sigma_1(A)^2 + \sigma_1(B)^2 - \sqrt{\sigma_1(A)^4 + \sigma_{r+1}(Z)^2} \geq 0$. We have $\sigma_1(B)^2 \geq \sigma_1(Z)^2/\sigma_1(A)^2$. Then

$$\begin{aligned} \sigma_1(A)^2 &+ \sigma_1(B)^2 - \sqrt{\sigma_1(A)^4 + \sigma_{r+1}(Z)^2} \\ &\geq \sigma_1(A)^2 + \sigma_1(Z)^2/\sigma_1(A)^2 - \sqrt{\sigma_1(A)^4 + \sigma_{r+1}(Z)^2} \\ &\geq \sigma_1(A)^2 + \sigma_{r+1}(Z)^2/\sigma_1(A)^2 - \sqrt{\sigma_1(A)^4 + \sigma_{r+1}(Z)^2} \\ &= (\sigma_1(A)^4 + \sigma_{r+1}(Z)^2)\left(\frac{1}{\sigma_1(A)^2} - \frac{1}{\sqrt{\sigma_1(A)^4 + \sigma_{r+1}(Z)^2}}\right) \\ &\geq 0, \end{aligned}$$

which proves the result.

**Smallest non-zero eigenvalue of the Hessian.** We have $\lambda_{\min\neq0}(H) = \lambda_{r(n+m)-r^2}(H)$, as the Kernel of $H$ has dimension $r^2$. Equation 8 shows that all the eigenvalues of $H_1$ are also eigenvalues of

$H_2$. Therefore, $\lambda_{r(n+m)-r^2}(H) \leq \min(\sigma_r(A)^2, \sigma_r(B)^2)$. For the lower bound, we apply the Weyl inequality:

$$\begin{aligned}
\lambda_{r(n+m)-r^2}(H) &= \lambda_{r(n+m)-r^2}(H_1 + H_2) \\
&\geq \lambda_{r(n+m)-r^2}(H_1) + \lambda_{r(n+m)}(H_2) \\
&= \min(\sigma_r(A)^2, \sigma_r(B)^2) - \sigma_{r+1}(Z),
\end{aligned}$$

according to Lemma A.2.

When the minimizer is balanced, it holds $\sigma_r(A)^2 = \sigma_r(B)^2 = \sigma_r(Z)$. This is the maximal value for the lower bound. Indeed, let $(A, B)$ be any minimizer of the loss $f$, i.e., $AB = LR_r(Z)$. Denote $U\Sigma V^\top$ the thin SVD of $LR_r(Z)$, i.e., with $\Sigma \succ 0$ of size $r \times r$. We can write $A = U\Sigma^{1/2}P$, $B = P^{-1}\Sigma^{1/2}V^\top$ for some invertible matrix $P \in GL_r(\mathbb{R})$. Then $\sigma_r(Z) = \sigma_r(U\Sigma V^\top) = \sigma_r(\Sigma^{1/2}PP^{-1}\Sigma^{1/2}) \geq \sigma_r(\Sigma^{1/2}P)\sigma_r(P^{-1}\Sigma^{1/2})$ by the Weyl inequality. Hence, $\sigma_r(Z) \geq \sigma_r(A)\sigma_r(B)$. We have proven that balanced minimizers maximize the lower bound $\min(\sigma_r(A)^2, \sigma_r(B)^2) - \sigma_{r+1}(Z)$. Now, it is easy to check that when $(A, B)$ is balanced, the vector $\begin{pmatrix} o_r \otimes u_{r+1} \\ v_{r+1} \otimes o_r \end{pmatrix}$ with

- $o_r$ an eigenvector of $A^\top A = BB^\top$ associated with eigenvalue $\sigma_r(Z)$,
- $u_{r+1}$ the column $r + 1$ of $U$,
- $v_{r+1}$ the column $r + 1$ of $V$,

is an eigenvector of $H(A, B)$ with the eigenvalue $\sigma_r(Z) - \sigma_{r+1}(Z)$, which proves that $\lambda_{\min \neq 0}(H(A, B)) = \sigma_r(Z) - \sigma_{r+1}(Z)$ when $(A, B)$ is balanced.

## B   STRUCTURE OF THE HYPERBALANCED MANIFOLD $\mathcal{H}$

The following proposition further details the structure of the set $\mathcal{H}$.

**Proposition B.1** (Equivalent descriptions of $\mathcal{H}$). *The set $\mathcal{H}$ is a smooth manifold in a neighborhood of full-rank points, with dimension equal to that of the rank-$r$ manifold $\mathcal{N}_r := \{X \in \mathbb{R}^{n \times m} : \operatorname{rank}(X) \leq r\}$. The product mapping $(A, B) \mapsto AB$ is a surjective map from $\mathcal{H}$ onto $\mathcal{N}_r$. If one locally fixes a consistent sign convention for the singular vectors in an SVD decomposition $X = USV^\top$, then the mapping $X \mapsto (US^{1/2}, S^{1/2}V^\top)$ defines a smooth local inverse at points $X$ with non-repeated singular values. Equivalently, $\mathcal{H}$ admits the explicit description*

$$\mathcal{H} = \{(US^{1/2}, S^{1/2}V^\top) : U^\top U = V^\top V = I_r, \; S \in \mathbb{D}_+^r\}. \tag{9}$$

*Proof.* We prove Equation (5). ($\subseteq$) Take $(A, B) \in \mathcal{H}$ with $A^\top A = BB^\top = S \in \mathbb{D}_+^r$. Set $U := AS^{-1/2}$ and $V := B^\top S^{-1/2}$. Then $U^\top U = I_r, V^\top V = I_r$, and $A = US^{1/2}, B = S^{1/2}V^\top$. ($\supseteq$) Conversely, if $A = US^{1/2}$ and $B = S^{1/2}V^\top$ with $U^\top U = V^\top V = I_r$ and $S \in \mathbb{D}_+^r$, then $A^\top A = S$ and $BB^\top = S$, so $(A, B) \in \mathcal{H}$. $\qquad\square$

**Remark B.1** (Smoothness of $P$). *It is important to note that the definition of $P$ in Equation (6) is not entirely unambiguous: singular vectors are determined only up to sign, and in the presence of repeated singular values they are even invariant under rotations within the degenerate subspace. When analyzing the convergence of optimization schemes over $X = AB$, this ambiguity is harmless, as discussed in Section 3.2. However, to guarantee that $P$ is smooth, one must restrict attention to points $X$ with distinct singular values and adopt a locally consistent sign convention for the singular vectors.*

With Remark B.1 in mind, the next proposition shows that $P$ locally enables the definition of a retraction map (Absil & Malick (2012)) that preserves the product $AB$.

**Proposition B.2** (Properties of $P$). *Let $P$ be as in Equation (6). Then $P(A, B) \in \mathcal{H}$, and if $(A, B) \in \mathcal{H}$, then $P(A, B) = (A, B)$. Furthermore, $P$ preserves the product $\Pi(A, B) := AB$, i.e., $\Pi(P(A, B)) = \Pi(A, B)$. The map $P$ acts locally as a first–order retraction on $\mathcal{H}$: for any $Z := (A, B) \in \mathcal{H}$ such that $AB$ has distinct singular values, and for any $\Delta \in T_Z\mathcal{H}$ in the*

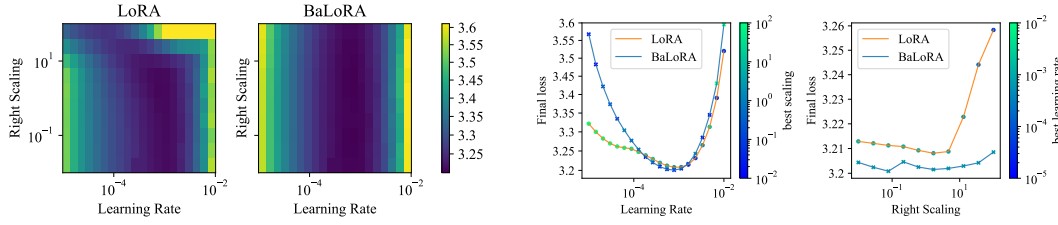

(a) Final test loss vs. (learning rate, scaling)

(b) Best test loss per learning rate or scaling

Figure 5: **LLM finetuning results, uncropped.** Final test loss of LoRA and BaLoRA for GPT-2 finetuning on Wikitext-2-raw-v1. This figure plots the same data as in Figure 3, but for the entire range of tested learning rates and showing the entire range of the final loss over the grid.

*tangent plane of $\mathcal{H}$ at $Z$, the map $(Z, \Delta) \mapsto P(Z + \Delta)$ defines a first-order retraction, namely $P(Z + \Delta) = Z + \Delta + o(\Delta)$.*

*Proof.* Fix $(A, B) \in \mathcal{H}$ and assume $Z := AB$ has distinct singular values. By standard perturbation theory for the SVD (with a consistent choice of signs), the reduced SVD $Z \mapsto (U, S, V)$ depends $C^1$-smoothly on $Z$ in a neighborhood of $Z$, hence the map

$$P(A', B') = \left(U(A'B')\, S(A'B')^{1/2},\ S(A'B')^{1/2} V(A'B')^{\top}\right)$$

is $C^1$ in $(A', B')$ near $(A, B)$. Moreover, $P$ fixes $\mathcal{H}$: if $(A', B') \in \mathcal{H}$ then $P(A', B') = (A', B')$.

Let $\Delta \in T_{(A,B)}\mathcal{H}$. By the definition of the tangent space of an embedded submanifold, there exists a $C^1$ curve $\gamma : (-\epsilon, \epsilon) \to \mathcal{H}$ with $\gamma(0) = (A, B)$ and $\dot{\gamma}(0) = \Delta$. Since $P$ fixes $\mathcal{H}$ pointwise, $P(\gamma(t)) = \gamma(t)$ for all $t$ small. Differentiating at $t = 0$ and using the chain rule yields

$$DP(A, B)[\Delta] = \frac{d}{dt}\Big|_{t=0} P(\gamma(t)) = \frac{d}{dt}\Big|_{t=0} \gamma(t) = \Delta.$$

Because $P$ is $C^1$, its first-order expansion at $(A, B)$ gives, for any $\varepsilon \to 0$,

$$P\big((A, B) + \varepsilon\Delta\big) = P(A, B) + \varepsilon\, DP(A, B)[\Delta] + o(\varepsilon) = (A, B) + \varepsilon\Delta + o(\varepsilon).$$

$\square$

## C    ADDITIONAL EMPIRICAL RESULTS

We provide in this section some additional figures.

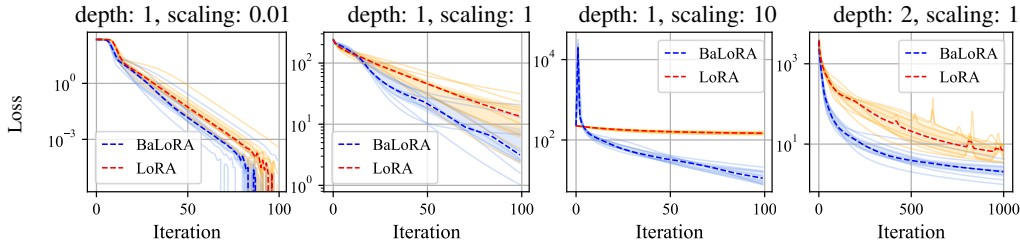

Figure 6: Evolution of the loss of LoRA with GD versus GD-BaLoRA for four different problems. The dotted lines are the median of 8 curves for each method, each curve corresponding to a different seed for the initialization, while the target and the learning rate are kept fixed. Both methods use the traditional LoRA init, with $A_0 = 0$, $B_0$ random Gaussian, and a scaling $\alpha/r$. The first three plots corresponds to a square one-layer linear network of size 20, with a lora rank of 4, and $\alpha \in \{0.01r, r, 10r\}$. The fourth plot was obtained with a two-layer linear network of size 20, a lora rank of 4, and $\alpha = r$. BaLoRA converges faster in all situations, with a particular advantage when the scaling $\alpha$ is large (third plot), where LoRA fails to converge.

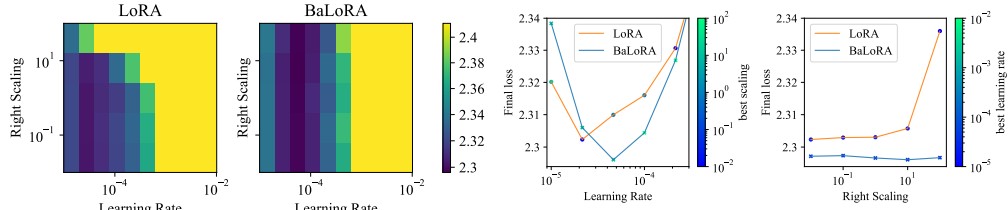

Figure 7: Final test loss of LoRA and BaLoRA for a grid of learning rates and initialization scalings, when finetuning Llama-3.2 on Wikitext-2-raw-v1. The left plots represent this final test loss for each pair (learning rate, initialization scaling), while the right plots only show the best test loss for each learning rate (resp. initialization scaling), i.e., the test loss corresponding to the best choice of initialization scaling (resp. learning rate), independently for the two curves. The color of points indicates the magnitude of this optimal initialization scaling (resp. learning rate): darker points correspond to smaller values. While the optimal learning rate is approximately constant across scalings for both methods, we observe that BaLoRA tends to prefer larger scalings than LoRA.

# D  LLM USAGE

The authors of this paper used Large Language Models to aid and polish the writing of this paper, for finding related work, and as a tool to make some of the proofs.

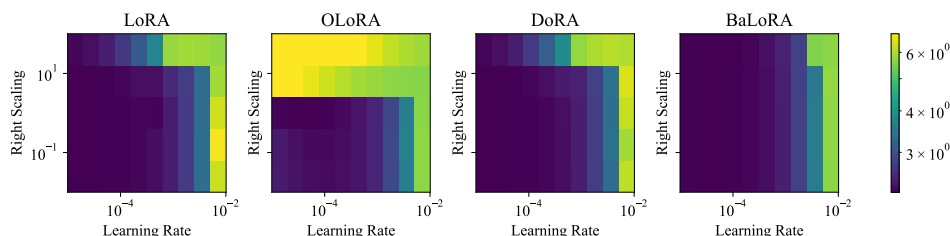

Figure 8: Final test loss of LoRA, OLoRA, DoRA and BaLoRA for a grid of learning rates and initialization scalings, when finetuning Llama-3.2 on Wikitext-2-raw-v1. The plot represents the final test loss for each pair (learning rate, initialization scaling). We observe that BaLoRA is more robust to hyperparameter changes than LoRA, OLoRA and DoRA, and achieves a similar loss value for the best choice of hyperparameters.

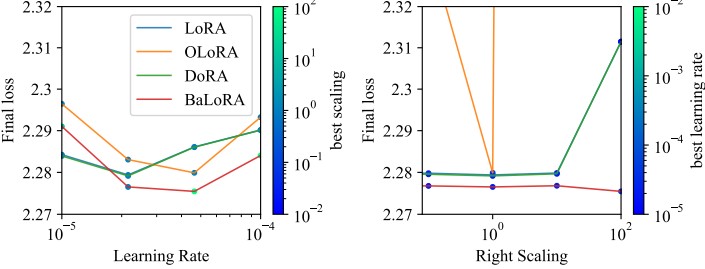

Figure 9: Final test loss of LoRA, OLoRA, DoRA and BaLoRA for a grid of learning rates and initialization scalings, when finetuning Llama-3.2 on Wikitext-2-raw-v1. The left (resp. right) plot shows the best test loss for each learning rate (resp. initialization scaling), i.e., the test loss corresponding to the best choice of initialization scaling (resp. learning rate), independently for the two curves. The color of points indicates the magnitude of this optimal initialization scaling (resp. learning rate): darker points correspond to smaller values. We observe that BaLoRA achieves a smaller test loss than the other methods for the best choice of hyperparameters, and is more robust to scaling variations.

