# OpenReview forum: "Balanced Low-Rank Adaptation: Removing Invariance for Fast and Stable Fine-Tuning"
_ICLR.cc/2026/Conference — Submitted to ICLR 2026_

### Official Review · Reviewer_dkCH · 2025-10-28

**Soundness:** 3
**Presentation:** 2
**Contribution:** 2
**Rating:** 4
**Confidence:** 3

**Summary:**

The paper proposes BaLoRA, a method that enforces a balance constraint between LoRA’s low-rank factors by projecting them onto a hyperbalanced manifold after each optimizer step. This aims to improve conditioning and accelerate convergence. The authors provide theoretical analysis for linear layer and empirical results on GPT-2 and LLaMA-3B, showing moderate improvements.

**Strengths:**

- This paper provides a novel perspective on the traing dynamics of LoRA from the condition number of minimizers.
- The mathematical derivations are clearly presented and easy to follow.

**Weaknesses:**

- W1. The toy example is overly simplified and cannot adequately capture how conditioning affects LoRA’s convergence. A single-layer linear network only reflects matrix factorization behavior and fails to model the impact of activation functions or deeper architectures.

 - W2. In Section 3.2, the authors state that BaLoRA with Adam is a heuristic and restricts theoretical analysis to the gradient descent variant. However, the paper still presents BaLoRA (with Adam) as the main method—even in the abstract—thus weakening the theoretical soundness of the overall claims.

 - W3. The empirical improvements in test loss are marginal compared to vanilla LoRA. The paper would benefit from evaluating on more diverse benchmarks (e.g., math or code tasks) and reporting accuracy as well as standard deviation over multiple runs.

**Questions:**

- The authors make a strong claim in Line 124 that “previous studies in optimization of LoRA do not address fine-tuning of machine learning models.” Could the authors clarify or justify this statement?

---

> ### Author Response · Authors · 2025-11-26
>
> We thank Reviewer dkCH for their comments, and for noting that our paper offers a **novel perspective** on the training dynamics of LoRA, with **clearly presented** mathematical derivations.
>
> > W1. The toy example is overly simplified and cannot adequately capture how conditioning affects LoRA’s convergence. A single-layer linear network only reflects matrix factorization behavior and fails to model the impact of activation functions or deeper architectures.
>
> As a first remark, while our analysis focuses on the loss $\lVert AB - Y\rVert^2$, the same approach yields bounds for losses of the form $\lVert S A B T - Y\rVert^2$ for fixed matrices $S$ and $T$. Our bound is impacted by the conditioning of the linear map $(A,B) \mapsto (SA, BT)$, which is controlled by $\frac{\max\bigl(\sigma_{\max}(S), \sigma_{\max}(T)\bigr)^{2}}
> {\min\bigl(\sigma_{\min}(S), \sigma_{\min}(T)\bigr)^{2}}$. This setting makes it possible to address fine tuning of an arbitrary layer in a **deep linear network**.  We will clarify this point in the final version of the paper.
>
> We fully acknowledge that analyzing the conditioning of deep non-linear networks remains far beyond current theoretical tools. That said, before our work no rigorous study of conditioning was available even in the simplest case of matrix factorization. More precisely, *all existing theoretical analyses* of LoRA focus exclusively on the one-layer linear case (Ye & Du 2021, Ghosh et al. 2025, Nguegnang et al. 2024). Moreover, the only research paper that studies conditioning (Ghosh et al. 2025) considers  *matrix factorization* under *strong assumptions* on the structure of the minimizers. Our results thus extends this line of research by analyzing the limiting conditioning of LoRA and BaLoRA in a broader class of linear models, which is a necessary step before tackling deeper or non-linear architectures.
>
> To conclude, while we understand the concern raised by the reviewer, we believe it is unrealistic at this stage to expect strong theoretical guarantees for non-linear networks. Even in the simple case of a single nonlinearity, the Hessian of the loss contains complicated dependencies between $A$ and $B$ which cannot be exactly captured by operator of the form $SABT$.
>
> > W2. In Section 3.2, the authors state that BaLoRA with Adam is a heuristic and restricts theoretical analysis to the gradient descent variant. However, the paper still presents BaLoRA (with Adam) as the main method—even in the abstract—thus weakening the theoretical soundness of the overall claims.
>
> There is indeed a difference between the theoretically analyzed variant (BaLoRA with gradient descent) and the practical method we highlight in the abstract (BaLoRA with Adam). This gap between theory and practice is hard to avoid: Adam is among the most widely used and effective optimizers for modern neural networks, including LoRA-based fine-tuning of LLMs, but its theoretical analysis remains challenging and scarce. Even on convex quadratic objectives, sign gradient descent (an idealized version of Adam) can fail to converge  [(Karimireddy et al 2019)](https://arxiv.org/abs/1901.09847).
>
> Note however that our analysis can cope with SGD. For simplicity, assume that the variance of the noisy gradient at each step is uniformly bounded by $\sigma^{2}$. Then, with a fixed stepsize $\tau$, the SGD iterates contract in expectation towards a ball of radius $\tau \sigma^{2}$ around the local minimizer, with a contraction factor determined by the conditioning of the Hessian (Bottou et al, SIAM review 2028).  We will update the final version of the manuscript with these clarifications.
>
> Still, our goal was to leverage our theoretical insights on a simplified setting to propose a methodology which, although not entirely conssitent with our theoretical assumptions, provides some practical benefits as compared to LoRA. Extending our analysis to include optimizers used in practice, such as Adam, is a very challenging mathematical problem.
>
>
> > W3. The empirical improvements in test loss are marginal compared to vanilla LoRA. The paper would benefit from evaluating on more diverse benchmarks (e.g., math or code tasks) and reporting accuracy as well as standard deviation over multiple runs.
>
> We have added two recognizer PEFT baselines, DoRA and OLoRA, to our benchmark of Llama-3.2-3B on Wikitext-2-raw-v1. The results appear in Figures 8 and 9 of the revised manuscript, in Appendix C. BaLoRA beats DoRA and OLoRA in this setting, and is more robust to scaling variations.
>
> We acknowledge the need for more diverse benchmarks, and averaging over multiple runs.

---

> ### Author Response · Authors · 2025-11-26
>
> > Q1. The authors make a strong claim in Line 124 that “previous studies in optimization of LoRA do not address fine-tuning of machine learning models.” Could the authors clarify or justify this statement?
>
> We were unable to find the exact sentence cited by the reviewer in our submission, but we presume the reviewer refers to the following: *"While matrix factorization can be viewed as a special case of LoRA, these studies do not address fine-tuning of machine learning models."*
>
> This statement highlights that, while LoRA can mathematically be related to matrix factorization, classical matrix factorization studies do not address the primary purpose of LoRA, that is fine-tuning pretrained, large models whose weight matrices are **full-rank**. Hence, we did not intend this as a strong claim, but rather as a way to position our contributions in the context of matrix factorization. We will reformulate this statement in the revised version to remove any potential ambiguity.

---

> > ### Comment · Reviewer_dkCH · 2025-11-27
> >
> > I acknowledge that I have read the authors’ response. Although the authors claim that the new setting generalizes to arbitrary deep linear networks, it still differs substantially from the nonlinear scenarios encountered in practical neural networks, which weakens the relevance of the analysis. I will maintain my score.

---

> > > ### Author Response · Authors · 2025-11-29
> > >
> > > We understand the concern raised by the reviewer, but we believe that the assessment is overly harsh and based on expectations that are, at present, unrealistic. To the best of our knowledge, the problem of deriving sharp, explicit bounds on the conditioning of the fine-tuning landscape remains well beyond the current frontier of the mathematical theory of deep learning.
> > >
> > > On the positive side, it is possible to bound the conditioning of the fine-tuning problem for a *single* adapter $\theta := (A,B)$ by the product of the conditioning of the underlying network and the bounds we derive. More precisely, let $\varphi$ denotes the map encapsulating all layers after the fine-tuned one, $X$ denotes the pre-activations,
> > > $W$ the fixed trained weights, the residual is $r := \varphi(Z) - Y$ with $Z := (W+AB) X$.
> > > The fine-tuning loss is  $f(\theta) := \|r\|^{2}$
> > > Then the Hessian of $f$ with respect to $\theta$ is
> > > $$\partial^{2} f(\theta)
> > >     = J^{\top} [\partial^{2} m(\theta)] J
> > >       + \partial^{2}\varphi(Z)[r,r],$$
> > > where $m(\theta) := AB$, $J := \partial\varphi(Z)$ is the Jacobian.
> > >
> > > The two cases where one can leverage our bounds because the second term vanish are:
> > > - **ReLU or Leaky ReLU activations** because $\partial^{2}\varphi = 0$,
> > > - **Interpolation regime** because  $r = 0$.
> > >
> > > In either of these regimes, the Hessian reduces to $\partial^{2} f(\theta) = J^{\top}\bigl[\partial^{2} m(\theta)\bigr] J$, and the conditioning is bounded by  $\kappa\bigl(\partial^{2} f(\theta)\bigr) \leq \kappa(J)^{2} \kappa(\partial^{2} m(\theta))$, and our bounds precisely control $\kappa(\partial^{2} m(\theta))$.
> > >
> > >
> > > We will include this observation in the final version of the paper, in the form of a short discussion on the deeper and nonlinear case. This bound is admittedly crude, but if this is the type of result the reviewer had in mind, we kindly ask that they reconsider their assessment in light of the above clarification.

---

### Official Review · Reviewer_yGgB · 2025-10-31

**Soundness:** 2
**Presentation:** 3
**Contribution:** 2
**Rating:** 4
**Confidence:** 3

**Summary:**

This paper introduces Balanced Low-Rank Adaptation (BaLoRA), a variant of LoRA designed to improve conditioning and convergence during fine-tuning of large language models. The core idea is to project LoRA adapters onto a balanced manifold to ensure the optimization dynamics converge toward better-conditioned minima without changing the effective update. The authors provide theoretical analysis of LoRA’s conditioning, prove that balanced minimizers yield optimal condition numbers, and derive efficient projection algorithms. Empirical results on demonstrate faster convergence and the robustness to hyperparameter choices.

**Strengths:**

Overall, I think this paper provides a novel geometric insight into LoRA conditioning. This contribution is quite significant, as the paper effectively bridges theoretical motivation with simple yet elegant algorithmic design. The empirical results align well with the theoretical findings and further demonstrate the robustness of BaLoRA.

**Weaknesses:**

* I think the theoretical scope is a little bit limited. The analysis is mostly confined to one-layer linear networks and has limited applicability to transformer layers. Moreover, although the paper aims to improve the convergence, the convergence analysis under Adam and SGD is not fully addressed, especially for those practical nonconvex cases. I think this is not a major drawback of this paper.

* The experimental settings and results are somewhat limited. For example, the authors could compare with the scale AdamW in paper [1], OLoRA [2], and [3]. Moreover, only the loss is reported; the missing downstream task accuracy should be very important in studying LoRA-based methods.

[1] Zhang, Fangzhao, and Mert Pilanci. "Riemannian preconditioned lora for fine-tuning foundation models." arXiv preprint arXiv:2402.02347 (2024).

[2] Kerim Buyukakyuz. Olora: Orthonormal low-rank adaptation of large language models. arXiv
preprint arXiv:2406.01775, 2024.

[3] Juneyoung Park, Minjae Kang, Seongbae Lee, Haegang Lee, Seongwan Kim, and Jaeho Lee. Riemannian optimization for lora on the stiefel manifold. arXiv preprint arXiv:2508.17901, 2025.

* Does the balanced projection happen every step? Did the author also consider the less frequent balanced projection case? Moreover, how do the authors choose the learning rates? By simple grid search or some other methods? I am asking this since 1) From Figure 3, I think either training loss or test loss is not that sensitive to the detailed learning rate with a similar scalar. 2) The learning rate adopted in some experiments is like 3.8e-4, which is less common in other LoRA papers. So naturally, my next question is, can you also provide the learning rate sensitivity in downstream tasks?

**Questions:**

See detailed questions in the weaknesses part.

---

> ### Author Response · Authors · 2025-11-26
>
> We thank Reviewer yGgB for their feedback. We are glad they found that our paper provides a **novel geometric insight** into LoRA conditioning, makes a **significant** contribution, and **effectively bridges theoretical motivation with simple yet elegant algorithmic design**.
>
> > I think the theoretical scope is a little bit limited. The analysis is mostly confined to one-layer linear networks and has limited applicability to transformer layers.
>
> As a first remark, while our analysis focuses on the <!--matrix factorization--> loss $\lVert AB - Y\rVert^2$, the same approach yields bounds for losses of the form $\lVert S A B T - Y\rVert^2$ for fixed matrices $S$ and $T$. Our bound is impacted by the conditioning of the linear map $(A,B) \mapsto (SA, BT)$, which is controlled by $\frac{\max\bigl(\sigma_{\max}(S), \sigma_{\max}(T)\bigr)^{2}}
> {\min\bigl(\sigma_{\min}(S), \sigma_{\min}(T)\bigr)^{2}}$. This setting makes it possible to address fine tuning of an arbitrary layer in a **deep linear network**.  We will clarify this point in the final version of the paper.
>
> We fully acknowledge that analyzing the conditioning of deep non-linear networks remains far beyond current theoretical tools. That said, before our work no rigorous study of conditioning was available even in the simplest case of matrix factorization. More precisely, *all existing theoretical analyses* of LoRA focus exclusively on the one-layer linear case (Ye & Du 2021, Ghosh et al. 2025, Nguegnang et al. 2024). Moreover, the only research paper that studies conditioning (Ghosh et al. 2025) considers  *matrix factorization* under *strong assumptions* on the structure of the minimizers. Our results thus extends this line of research by analyzing the limiting conditioning of LoRA and BaLoRA in a broader class of linear models, which is a necessary step before tackling deeper or non-linear architectures.
>
> To conclude, while we understand the concern raised by the reviewer, we believe it is unrealistic at this stage to expect strong theoretical guarantees for non-linear networks. Even in the simple case of a single nonlinearity, the Hessian of the loss contains complicated dependencies between $A$ and $B$ which cannot be exactly captured by operator of the form $SABT$.
>
>
> > Moreover, although the paper aims to improve the convergence, the convergence analysis under Adam and SGD is not fully addressed, especially for those practical nonconvex cases. I think this is not a major drawback of this paper.
>
> This is a fair point: our theoretical guarantees cover only a local convergence analysis for gradient descent. The same reasoning applies to SGD. For simplicity, assume that the variance of the noisy gradient at each step is uniformly bounded by $\sigma^{2}$. Then, with a fixed stepsize $\tau$, the SGD iterates contract in expectation toward a ball of radius $\tau \sigma^{2}$ around the local minimizer, with a contraction factor determined by the conditioning of the Hessian (Bottou et al, SIAM review 2028). In contrast, addressing Adam, which rescales gradient updates, remains far out of reach. Even on convex quadratic objectives, sign gradient descent (an idealized version of Adam) can fail to converge  [(Karimireddy et al 2019)](https://arxiv.org/abs/1901.09847). We will update the final version of the manuscript with these clarifications.

---

> ### Author Response · Authors · 2025-11-26
>
> > The experimental settings and results are somewhat limited. For example, the authors could compare with the scale AdamW in paper [1], OLoRA [2], and [3].
>
> We have added two recognized baselines, DoRA and OLoRA, to our benchmark of Llama-3.2-3B on Wikitext-2-raw-v1. The results appear in Figures 8 and 9 of the revised manuscript, in Appendix C. BaLoRA beats DoRA and OLoRA in this setting, and is more robust to scaling variations.
>
> > Moreover, only the loss is reported; the missing downstream task accuracy should be very important in studying LoRA-based methods.
>
> The lack of downstream task evaluations is a valid point. A fair downstream comparison would require substantial additional work, in particular a full grid search over hyperparameters such as step sizes and initialization scales. Conducting this thoroughly goes beyond the scope of the current paper, whose aim is to stay task-agnostic and to focus on the optimization properties of the method itself. We will clarify this positioning in the revised version and highlight downstream evaluation as a natural next step.
>
> > Does the balanced projection happen every step? Did the author also consider the less frequent balanced projection case?
>
> Indeed, in our initial implementation, the balanced projection is applied at every step. We had not considered less frequent projections, which is an interesting follow-up question. To explore this point, we have added a hyperparameter to tune the frequency of the projection step. We will compare the performance of this variant (projecting every 10 iterations) with the previously described methods. We thank the reviewer for suggesting this methodological extension, which can be easily incorporated and provides additional insights on BaLoRA.
>
>
> > Moreover, how do the authors choose the learning rates? By simple grid search or some other methods? I am asking this since 1) From Figure 3, I think either training loss or test loss is not that sensitive to the detailed learning rate with a similar scalar.
>
> The learning rates are indeed chosen by simple grid search.
>
> We are unsure we fully understood the reviewer's interpretation of Figure 3 and would like to clarify the reported results. Figure 3(a) shows the final test loss value (the darker, the smaller) for each pair (learning rate, scaling). We observe that the final test loss value varies significantly along horizontal lines (*i.e.*, for a fixed scaling). In particular, the yellow color corresponds to *any loss value above* 3.47, and the actual values for very small ($10^{-5}$) and very large ($10^{-2}$) learning rates are above 3.6. We thresholded the colorbar to provide a finer picture of the loss variations around the optimal value. To clarify this point, we added a figure (Figure 5 in Appendix C) showing the whole range of loss values in the same setting as Figure 3.
>
> We hope this clarifies our results, and we would be happy to further address any remaining questions regarding Figure 3(a).
>
>
> > 2) The learning rate adopted in some experiments is like 3.8e-4, which is less common in other LoRA papers.
>
> The value 3.8e-4 is the learning rate that minimized the test loss for both LoRA and BaLoRA when fine-tuning GPT-2 on Wikitext, which is why we used it in Figure 4.

---

### Official Review · Reviewer_TXuS · 2025-11-06

**Soundness:** 2
**Presentation:** 3
**Contribution:** 2
**Rating:** 4
**Confidence:** 4

**Summary:**

This paper proposes BaLoRA, a more effective approach to fine-tuning large language models.
LoRA is a popular method that adds small, trainable matrices to a large model; however, it can be unstable to train. BaLoRA addresses this by maintaining the balance of those small matrices after every training step.
This balancing makes training faster, more stable.
In tests on models like GPT-2 and Llama-3, BaLoRA trained more quickly and achieved lower loss than LoRA.

**Strengths:**

- This paper attempts to mathematically manipulate LoRA to achieve better convergence.
- It demonstrates that using balanced LoRA could improve convergence.
- The theoretical analysis for convergence is quite interesting.
- The explanations and writing are clear and easy to follow.

**Weaknesses:**

- There is a lack of novelty in this work; many studies have already explored SVD LoRA (TriLoRA, OPLoRA, ...), and I don't see significant improvements or differences here beyond the theoretical aspect.
- Why the only metric used is loss? I think that loss doesn't provide much information about the performance of LLMs. There are several better metrics available to evaluate performance (MTbanch, vicuna, ...).
- Why is the only baseline used here appears to be LoRA? Since there are many variations of LoRA, including those related to SVD (TriLoRA, OPLoRA, ...), it would have been beneficial to compare the results against these other baselines, not just LoRA.

**Questions:**

- Why is LoRA the only baseline method used? It would be helpful to include additional baseline methods.
- If you could clarify how this work is novel compared to other SVD-LoRA methods, I would be willing to reconsider my score.
- The performance improvement would be more evident if better evaluation metrics (such as MTbench or Vincuna) were used, rather than just loss values.

---

> ### Author Response · Authors · 2025-11-26
>
> We thank Reviewer TXuS for their questions and comments, and for finding our theoretical analysis **interesting** and **clear**.
>
>
> > There is a lack of novelty in this work; many studies have already explored SVD LoRA (TriLoRA, OPLoRA, ...), and I don't see significant improvements or differences here beyond the theoretical aspect. If you could clarify how this work is novel compared to other SVD-LoRA methods, I would be willing to reconsider my score.
>
> The reviewer is correct that some LoRA variants rely on SVD, but there is a clear difference between such approaches and our method. We appreciate the opportunity to clarify how our contributions differ, and will include the following discussion in our paper:
> * OPLoRA (Xiong and Xie, 2025) appeared on arXiv several weeks after the ICLR deadline, thus we were not aware of this approach when submitting our work. This method performs SVD on the **pretrained weight matrix** (not on the low-rank adapter, as in BaLoRA), and sandwiches the low-rank adapter $AB$ between two fixed projection matrices. This does not enforce the pair $(A, B)$ to be balanced, and is thus fundamentally different from BaLoRA.
> * TriLoRA writes the low-rank adapter as $U\Sigma V^\top$, without enforcing orthogonality on $U$ and $V$ (see their explanations at the end of Section 3.3). Hence, this amounts to optimizing the loss over **three matrices** ($U, \Sigma, V$) instead of the **two matrices** $A, B$ used in LoRA and BaLoRA. Therefore, TriLoRA uses a different loss $\ell(U, \Sigma, V)$, which induces different dynamics. Moreover, TriLoRA does not enforce any kind of *balancedness* of the low-rank adapters.
> * Beyond these two methods, to our knowledge, no prior work implements the same idea as BaLoRA. If the reviewer had other references in mind, we would be grateful to know them so that we can position our contributions accordingly.
>
>
> > Why the only metric used is loss? I think that loss doesn't provide much information about the performance of LLMs. There are several better metrics available to evaluate performance (MTbanch, vicuna, ...).
>
> The lack of downstream task evaluations is a valid point. A fair downstream comparison would require substantial additional work, in particular a full grid search over hyperparameters such as step sizes and initialization scales. Conducting this thoroughly goes beyond the scope of the current paper, whose aim is to stay task-agnostic and to focus on the optimization properties of the method itself. We will clarify this positioning in the revised version and highlight downstream evaluation as a natural next step.
>
> > Why is the only baseline used here appears to be LoRA? Since there are many variations of LoRA, including those related to SVD (TriLoRA, OPLoRA, ...), it would have been beneficial to compare the results against these other baselines, not just LoRA.
>
> We have added two recognized baselines, DoRA and OLoRA, to our benchmark of Llama-3.2-3B on Wikitext-2-raw-v1. The results appear in Figures 8 and 9 of the revised manuscript, in Appendix C. BaLoRA beats DoRA and OLoRA in this setting, and is more robust to scaling variations.

---

### Official Review · Reviewer_q79Q · 2025-11-10

**Soundness:** 2
**Presentation:** 2
**Contribution:** 2
**Rating:** 2
**Confidence:** 4

**Summary:**

This paper proposes BaLoRA (Balanced Low-Rank Adaptation), which projects the low-rank adapter onto a super-balanced manifold after each optimization step. This improves the condition number while preserving the adaptation matrix AB, with negligible computational overhead. Validated on text prediction tasks for GPT-2 and Llama-3.2-3B models, BaLoRA converges faster and demonstrates greater robustness to hyperparameters.

**Strengths:**

* S1: The theoretical analysis in Section 2 provides a clear derivation of the Hessian and condition numbers for LoRA in the matrix factorization and general cases, offering some insight into why balanced minimizers might lead to faster asymptotic convergence.
* S2: BaLoRA's projection step is computationally lightweight, making it easy to integrate into existing LoRA pipelines without significant overhead.
* S3: The empirical evaluation shows robustness to initialization scaling and learning rates, with BaLoRA outperforming LoRA in high-scaling regimes on both synthetic and real LLM fine-tuning tasks.

**Weaknesses:**

* W1: The theoretical contributions are limited to toy one/two-layer linear networks, which do not capture the complexities of deep transformers or non-linear activations, rendering the "optimal conditioning" claims speculative for practical PEFT scenarios.
* W2: Experiments are narrow: only two models (GPT-2, Llama-3.2-3B) on a single dataset, with no comparisons to recent LoRA variants like DoRA,  AdaLoRA, or OLoRA, and lacking downstream task evaluations to assess generalization.
* W3: The paper overlooks key related works on balanced optimization and Riemannian methods for low-rank manifolds, leading to overstated novelty.
* W4: The conclusions of condition number analysis for linear networks have not been specifically validated in nonlinear Transformers. The experiments did not quantify the specific contribution of condition number optimization to convergence acceleration, and the correlation between theoretical value and practical effectiveness lacks empirical support.

**Questions:**

* Q1: Insufficient Empirical Validation: The experiments are limited to toy settings (one-layer linear networks) and only two models (GPT-2 and Llama-3.2-3B) on a single dataset (Wikitext), with no comparisons to state-of-the-art PEFT variants like DoRA, AdaLoRA, or O-LoRA. In the absence of evaluations on diverse downstream tasks or larger models, the claims of faster convergence and robustness remain unconvincing.

* Q2: BaLoRA is essentially a minor modification of standard LoRA, relying heavily on existing matrix factorization techniques and balancing concepts from prior works. The projection onto a balanced manifold adds little new insight; thus, the paper's contributions are overstated.

---

> ### Author Response · Authors · 2025-11-26
>
> We thank Reviewer q79Q for their feedback, and for acknowledging that our theory is **clear**, provides **insights** on BaLoRA, and that BaLoRA is **lightweight**, **easy to integrate**, and more **robust** than LoRA.
>
> > W1: The theoretical contributions are limited to toy one/two-layer linear networks, which do not capture the complexities of deep transformers or non-linear activations.
>
> As a first remark, while our analysis focuses on the <!--matrix factorization--> loss $\lVert AB - Y\rVert^2$, the same approach yields bounds for losses of the form $\lVert S A B T - Y\rVert^2$ for fixed matrices $S$ and $T$. Our bound is impacted by the conditioning of the linear map $(A,B) \mapsto (SA, BT)$, which is controlled by $\frac{\max\bigl(\sigma_{\max}(S), \sigma_{\max}(T)\bigr)^{2}}
> {\min\bigl(\sigma_{\min}(S), \sigma_{\min}(T)\bigr)^{2}}$. This setting makes it possible to address fine tuning of an arbitrary layer in a **deep linear network**.  We will clarify this point in the final version of the paper.
>
> We fully acknowledge that analyzing the conditioning of deep non-linear networks remains far beyond current theoretical tools. That said, before our work no rigorous study of conditioning was available even in the simplest case of matrix factorization. More precisely, *all existing theoretical analyses* of LoRA focus exclusively on the one-layer linear case (Ye & Du 2021, Ghosh et al. 2025, Nguegnang et al. 2024). Moreover, the only research paper that studies conditioning (Ghosh et al. 2025) considers  *matrix factorization* under *strong assumptions* on the structure of the minimizers. Our results thus extends this line of research by analyzing the limiting conditioning of LoRA and BaLoRA in a broader class of linear models, which is a necessary step before tackling deeper or non-linear architectures.
>
> To conclude, while we understand the concern raised by the reviewer, we believe it is unrealistic at this stage to expect strong theoretical guarantees for non-linear networks. Even in the simple case of a single nonlinearity, the Hessian of the loss contains complicated dependencies between $A$ and $B$ which cannot be exactly captured by operator of the form $SABT$.
>
> > Q1 and W2: Experiments are narrow: only two models (GPT-2, Llama-3.2-3B) on a single dataset, with no comparisons to recent LoRA variants like DoRA, AdaLoRA, or OLoRA, and lacking downstream task evaluations to assess generalization.
>
> As suggested, we have added DoRA and OLoRA to our benchmark of Llama-3.2-3B on Wikitext-2-raw-v1. The results appear in Figures 8 and 9 of the revised manuscript, in Appendix C. BaLoRA beats DoRA and OLoRA in this setting, and is more robust to scaling variations.
>
> The lack of downstream task evaluations is a valid point. A fair downstream comparison would require substantial additional work, in particular a full grid search over hyperparameters such as step sizes and initialization scales. Conducting this thoroughly goes beyond the scope of the current paper, whose aim is to stay task-agnostic and to focus on the optimization properties of the method itself. We will clarify this positioning in the revised version and highlight downstream evaluation as a natural next step.

---

> ### Author Response · Authors · 2025-11-26
>
> > W3: The paper overlooks key related works on balanced optimization and Riemannian methods for low-rank manifolds, leading to overstated novelty.
> > Q2: BaLoRA is essentially a minor modification of standard LoRA, relying heavily on existing matrix factorization techniques and balancing concepts from prior works.
>
> If the reviewer has specific references on balanced optimization or Riemannian methods that implement a similar mechanism as BaLoRA, we would be grateful if they could point us to them so that we can discuss them appropriately.
>
> We are not aware of prior work that follows the same idea as BaLoRA. Existing PEFT approaches that use singular value decompositions do so for completely different purposes. For example, OLoRA uses SVD only to initialize the LoRA factors according to the principal directions of the pretrained weights, whereas in BaLoRA, the projection is applied at each training step and does not depend on the pretrained weights.
>
> > W4: The conclusions of condition number analysis for linear networks have not been specifically validated in nonlinear Transformers. The experiments did not quantify the specific contribution of condition number optimization to convergence acceleration, and the correlation between theoretical value and practical effectiveness lacks empirical support.
>
> We agree that tracking the conditioning of the loss would be insightful to connect our theoretical insights with practice. However, empirically estimating the smallest eigenvalue of the Hessian in transformers is a research problem on its own. Existing methods such as Lanczos-type iterative algorithms do not scale well enough to yield stable estimates of extreme eigenvalues in our setting: for such large models, the smallest eigenvalue is expected to be extremely close to zero and numerically unstable to estimate. We therefore leave a systematic investigation of conditioning in nonlinear architectures to future work.

---

### Author Response · Authors · 2025-12-01
**Recap for Area Chair**

Below we summarize the main concerns raised by the reviewers during the discussion period and how we addressed each of them, so the Area Chair can clearly assess the points of disagreement and the clarifications provided.

1. Scope and Relevance of the Theoretical Analysis

Reviewers (especially q79Q, yGgB, dkCH) noted that the theory focuses on one-layer linear models, raising doubts about its applicability to deep or nonlinear architectures such as Transformers. They questioned whether the conditioning claims transfer to practical PEFT settings, and some (dkCH) considered the setting too toy to be informative.

*Our response:*

Our analysis of the conditioning extends beyond matrix factorization, enabling treatment of arbritrary layers within **deep linear networks**, and even some **deep non-linear networks** (ReLU/Leaky ReLU, interpolation). In such cases, our bound involves the conditioning of subsequent layers of the network.

To our knowledge, **no prior work** provides conditioning analysis for LoRA or BaLoRA beyond the one-layer case. Existing studies either assume restrictive structural conditions or do not analyze conditioning at all.

2. Gap Between Theoretical BaLoRA (GD) and Practical BaLoRA (Adam)

Reviewer dkCH pointed out that we analyze BaLoRA under gradient descent, yet the practical method uses Adam, so theoretical justifications may not fully apply.

*Our response:*

This gap is **unavoidable**: Adam lacks convergence guarantees even for simple convex quadratic objectives.

Our analysis **does extend to SGD**, and we provide an expectation-contraction argument showing how conditioning controls convergence under bounded gradient noise.

The goal is to *use the theoretical insights to motivate the practical algorithm*, even if Adam itself is not theoretically tractable.

This clarification will be made explicit in the revised manuscript.

3. Limited Experimental Scope

All reviewers pointed out that the evaluation covers only GPT-2 and Llama-3.2-3B on Wikitext-2, includes no downstream tasks (MTBench, Vicuna, code/math tasks), omits several PEFT baselines (AdaLoRA, OLoRA, DoRA, Stiefel LoRA), and lacks multi-run variance reporting.

*Our response:*

We added DoRA and OLoRA as baselines for Llama-3.2-3B; BaLoRA **outperforms or matches them** and is more robust to scaling.

We **intentionally did not include downstream tasks**: fair comparisons require extensive hyperparameter sweeps (initial scales, learning rates, etc.). The paper is positioned around optimization behavior rather than task-specific evaluation; this will be clarified in the revision.

We acknowledge the value of broader evaluation and multi-run averages, but this is **beyond the scope** of the present submission.

4. Novelty Concerns Regarding SVD-Based LoRA Variants

Some reviewers (TXuS, q79Q) suggested BaLoRA may be a minor extension of LoRA, or that prior SVD-based LoRA variants (TriLoRA, OPLoRA, etc.) implement similar ideas. They also mentioned potentially missing related work on balanced optimization or Riemannian low-rank methods.

*Our response:*

We clarified key differences between BaLoRA and SVD-based LoRA variants:

OPLoRA (released after the submission) applies SVD to pretrained weights, not to the low-rank adapters, and does not enforce balancedness.

TriLoRA parameterizes adapters as $U\Sigma V^\top$ without orthogonality constraints, altering both the loss landscape and optimization dynamics; it does not enforce any balancing condition.

**No prior PEFT method performs iterative per-step balancing**, which is central to BaLoRA.

We requested pointers if any additional related work exists.

We emphasized that BaLoRA’s novelty lies in introducing a theoretically motivated balanced factorization that improves conditioning.

---

### Meta-Review · Area_Chair_ugvZ · 2026-01-05

**Summary:**

From the perspective of overparameterization, this manuscript theoretically investigates LoRA through an analysis of the condition number and then discusses asymptotic convergence. This analysis motivates a new LoRA variant, termed BaLoRA, which is further supported by numerical experiments.

All reviewers acknowledge that the paper provides new theoretical insights into LoRA. However, they also raise generally consistent concerns, which result in uniformly negative scores. During the rebuttal period, only one reviewer responded and maintained their original score.

The authors argue that the theoretical analysis in the one-layer setting is still meaningful, a point with which the AC agrees. However, under this setting, the numerical evaluation should be significantly strengthened, particularly comparisons with SVD-based LoRA methods that are closely related to condition number.

Overall, while the AC believes that this paper has the potential to make an impact. But in its current form, my final recommendation is REJECTION.

**Reviewer Concerns:**

All reviewers appreciated the theoretical analysis of LoRA from the perspective of the condition number (with the exception of Reviewer TXuS, who noted overlap with existing SVD-based LoRA methods). At the same time, they all agreed that the theoretical contributions are largely limited to one- or two-layer linear networks.

Regarding the empirical evaluation, all reviewers expressed dissatisfaction with the experimental design. In particular, the comparisons omit important LoRA variants, and reporting only training loss was considered insufficient and unacceptable (noted by Reviewers TXuS and yGgB).

**Reviewer Scores:**

The initial scores were 2/4/4/4. Only Reviewer dkCH provided additional comments during rebuttal and chose to maintain their score (4).

---

### Decision · Program_Chairs · 2026-01-26

Reject